# Analysis of Factors Influencing Vibration Suppression of Spray Boom-Air Suspension for Medium and Small-Scale High-Clearance Sprayers

**DOI:** 10.3390/s21206753

**Published:** 2021-10-12

**Authors:** Wei Qiu, Xinyue Yao, Yue Zhu, Hao Sun, Liangfu Zhou, Maohua Xiao

**Affiliations:** 1College of Engineering, Nanjing Agricultural University, Nanjing 210095, China; 9173010320@njau.edu.cn (X.Y.); zhuyue@njau.edu.cn (Y.Z.); 2020112023@stu.njau.edu.cn (H.S.); xiaomaohua@njau.edu.cn (M.X.); 2Engineering and Technical Training Center, Nanjing Vocational University of Industry Technology, Nanjing 210023, China; 2020101069@niit.edu.cn

**Keywords:** air suspension, dynamic characteristics, suspension parameters, vibration suppression, soil conditions

## Abstract

Medium and small-scale high-clearance sprayers are widely applied in medium and small-scale farms. Owing to power and load limitations, it is difficult to manage the complex system for suppressing spray boom vibration. This study was conducted to design a spray boom-air suspension system suitable for medium and small-size high-clearance sprayers by combining spray boom vibration suppression and the characteristics of air spring charging/discharging. Thus, this study aims to address the non-homogeneous distribution of spray triggered by the spray boom vibrations in medium and small high-clearance sprayers. The effects of different elastic elements on the vibration suppression effect of the spray boom were compared. According to the bench test, the dynamic response results of the spray boom under transient and sinusoidal excitations indicate that air spring is more conducive to vibration suppression than coil spring. The results obtained from the field experiments indicate that under the low solid soil condition, the spray boom air suspension should match a small additional air chamber with a volume of approximately 0.6 L, and the damping coefficient of the damper should be approximately 1792 N·s/m. In the case of the high firm soil, the spray boom air suspension should match a large additional air chamber with a volume of approximately 3.6 L, while the damping coefficient of the damper should be set to approximately 1316 N·s/m. The soil with low moisture content and high firmness are unfavorable to the vibration suppression of the spray boom. This study provides a reference for enhancing the vibration suppression of the spray boom-air suspension and improving the spray uniformity of the sprayer.

## 1. Introduction

High-clearance sprayers are commonly employed to protect plants in paddy fields because of their advantages of high ground clearance, narrow wheel track, and wide spray swath [1,2,3]. As a core feature of high-clearance sprayers, the sprayer deflects, oscillates, and rolls [4] under the excitation from the engine and the ground, which in turn influences spray uniformity [5,6], as well as the utilization rate of agrochemical chemicals [7]. Previous studies have shown that a suspension suppression system between the boom and sprayer can address the challenge of poor spraying triggered by the boom vibration [8,9,10].

Currently, the studies on spray boom suspension damping systems are mainly on large machines with spray amplitudes of 15–30 m [11,12,13,14,15,16]. The damping system usually composites of a combination of spring and damper, which can transform the rigid connection into a flexible tie, with low frequency, good vibration isolation performance, and creep resistance characteristics. Anthonis et al. [17,18], optimized the vertical suspension dynamics for a 39 m wide boom on a John Deere crop sprayer. Wu et al. [19], established the structure model of a 25 m wide spray boom with a spring-damper and optimized the ideal spring damping coefficient to achieve the vibration attenuation of the spray boom.

Cui et al. [20,21], designed a kind of dual-pendulum suspension adapted to 28 m spray boom, analyzed the influence of suspension structure parameters and control system parameters on the response characteristics of spray boom. In essence, they also realized the vibration attenuation of large spray rod by using spiral spring and damper. Some studies have paid attention to the vibration suppression of medium and short-length spray boom in recent years. For example, Zhuang et al. [22,23], designed a triangular structure suitable for the 18 m spray boom equipped with a spiral spring-damping system, and optimized the parameters combination of spring and damper for the vibration suppression.

However, the low frequency following the performance of the spring-damping system can lead to an imbalance of spray boom when the spray locomotive body wobbles on uneven road surface. To solve the above problems, Xue et al. [24], obtained the active suspension by adding a hydraulic cylinder to the two-link trapezoidal, and designed a set of complex vibration reduction systems under the adaptive fuzzy sliding mode control. Qiao et al. [25], proposed a control scheme for an active hydraulically activated suspension and proved that it performed significantly better than a spring-damping passive suspension. Although Xue and Qiao’s studies provided remarkable results, the systems were complex. Due to the limited power and load, a small-scale high-clearance sprayer (spray width of less than 12 m) has challenged to bear the complex spray boom damping systems.

The small-scale high-clearance sprayer has the characteristics of high mobility, which is suitable for small field operation and occupies a high quantity in South China and South Asia. The vibration suppression of spray boom is the key to restricting its application effect’s improvement. Kennes et al. [26], conducted a comparative analysis of experiments with three different structure types of 12 m boom on a hydraulic test rig and gave the optimization scheme. While components such as springs and dampers have not been used. Some researchers [27], use lighter, easier-to-install cables to attenuate the harmful movement of spray rods, but the cable only works in large deformation.

Air springs, which take air as the energy transmission medium, exhibit an optimal vibration damping effect, low cost, and easy control. They have been widely used in many fields [28,29]. In particular, with its small size the air spring can be easily installed under the spray boom of the small-scale high-clearance sprayer.

Based on the demand for the vibration suppression of medium and small high-clearance sprayers. Combined air spring characteristics, this study designed a spray boom air suspension system and analyzed the influence of elastic elements on the dynamic features of the spray boom- air suspension system and explored the impact of primary suspension parameters on the vibration suppression of this system under different soil conditions. Importantly, this study attempts to provide novel ideas for the vibration suppression of medium and small-scale high-clearance sprayers.

## 2. Materials and Methods

### 2.1. Spray Boom-Air Suspension System

The medium and small high-clearance sprayers consist of a spray system, spray boom-air suspension, boom lifting and folding mechanism, high-clearance chassis, and power distribution and transmission systems. The spray system comprises a medicine box, medicine pump, distributing valve, pressure gauge, sprinkler, medicine tube, and nozzle, which handles the atomization and spray of liquid medicine. The spray boom-air suspension includes a device for vibration damping and connection between the sprinkler and the chassis, which is consists of the boom, air spring, additional air chamber, air compressor, shock absorber, fixed support, and suspension base. The boom lifting and folding mechanism comprises a lifting electric linear actuator, left and right folding electric linear actuators, and a parallelogram mechanism. The design of the entire machine is illustrated in Figure 1, and its relevant parameters are presented in Table 1.

The sprinkler is fixed on the spray boom and moves with the boom, and the spray booms on both sides are independent of each other. The air spring and damper are arranged on each side, respectively. The end of the spray boom is hinged on the fixed support. The air spring and damper are connected between the spray boom and the suspension base in parallel, thereby functioning as support and vibration suppression simultaneously. The air spring is installed vertically and is connected with the additional air chamber, while the damper is installed obliquely, and it belongs to a mechanical multi-stage adjustable hydraulic type. During operation, the spray boom deflects, oscillates, and rolls under the influence of road excitation and engine vibration. The air spring deforms under force, and gas is exchanged between the main and auxiliary air chambers to generate damping force. The liquid inside the damper flows and generates damping force under the flow restriction of the orifice. Accordingly, the vibration energy of the spray boom-suspension system is suppressed, and the spray boom is gradually stabilized.

### 2.2. Parameter Design of Air Suspension for Spray Boom

The performance of air suspension predominantly depends on the spring and damper. The air spring generates elastic action by compressing the gas inside the airbag. The variable stiffness of the air spring can effectively limit the amplitude and optimize the natural frequency [30]. The damper attenuates the vibration by providing damping and suppresses the free vibration of the spring. The selection of the air spring and damper needs to be considered from several perspectives, such as load capacity, expansion stroke, etc.

Owing to the small horizontal deformation of the air spring during the entire process, which can be ignored, the spray boom vibration primarily involves the rotation around the pin shaft. Hence, the applied forces of the spray boom are simplified as illustrated in Figure 2. *O*, *A*, and *B* denote the centroid positions of the pin shaft, spring, and boom, respectively. In addition, *F_T_*, *F_N_*, and *G* represent the elastic force of the air spring, supporting force of the pin shaft, and gravity of the spray boom and connector, respectively. The dynamic load of the spring was calculated under the unbalanced force of the spray boom according to the mechanical theory.
(1)$$F_{T}=G+F_{N}+ma$$
(2)$$F_{T}L_{OA}=\left(G+ma\right)L_{OB}$$

Given that the mass of the spray boom and connector length are *m* =13 kg and $L_{OB}=2.5\;m$, the air spring is arranged on the 1-m long suspension base. To bear the load, the air spring should not be too close to the pin shaft, and to ensure sufficient installation space, the air spring is arranged at the 0.3-m horizontal distance from point *O* ($L_{OA}=0.3$ m). Without the suspension vibration isolation, the maximum motion acceleration range at the centroid was measured via the pre-test during the spray boom vibration, and the range was 23–25 $m/s^{2}$. $F_{N}=3317.6\;N$ when $a=25\;m/s^{2}$. The maximum load was $F_{Tmax}=3770\;N$. The bearing capacity of the air spring should be greater than $F_{Tmax}$.

When the maximum bearing capacity is applied to the suspension, the air spring must provide sufficient stiffness to attenuate the vibration. The stiffness calculation formula is expressed as [31]:(3)$$K=\frac{F_{Tmax}{\left(2\pi n\right)}^{2}}{g}$$
where gravity acceleration, *g* = 9.8 m/s^2^ and *n* denotes the natural frequency of suspension (Hz). According to the bench test, the first-order natural frequency of the spray boom is approximately 2.3 Hz. To improve the stability of the spray boom and weaken the resonance effect, the natural frequency of the air suspension should be significantly less than 2.3 Hz, which was set as *n* = 0.5–1.0 Hz. The suspension stiffness was obtained as *K* = 3796.78–15,187.10 N/m. When the design height of the selected air spring is 150 mm, the bearing capacity and stiffness should be greater than the aforementioned suspension design value [28].

The stroke of the air spring should meet the service requirements. When the machine was operating in the field without the installation of air suspension, the maximum deflection angle measured in the vertical plane was 6.2°~8° by the pre-experiment. To ensure that the air spring is not damaged, 8° was taken as the limit deflection angle of the spray boom to obtain the limit value of the change in air spring length. The spray boom rotates forward when the spring is extended, and reverses when the spring is compressed. The maximum length of the air spring was calculated according to the geometric relationship demonstrated in Figure 3:(4)$$L_{A1D1}=\sqrt{{\left(L_{A1B1}-L_{C1D1}\cos8{}^{\circ}\right)}^{2}+{\left(L_{B1C1}+L_{C1D1}\sin8{}^{\circ}\right)}^{2}}$$

The minimum length of the air spring was calculated:(5)$$L_{A2D2}=\sqrt{{\left(L_{A2B2}-L_{C2D2}\cos8{}^{\circ}\right)}^{2}+{\left(L_{B2C2}-L_{C2D2}\sin8{}^{\circ}\right)}^{2}}$$

Given that $L_{B1C1}=L_{B2C2}=150\;\mathrm{mm}$; hence,$L_{A1B1}=L_{A2B2}=L_{C1D1}=L_{C2D2}=300\;\mathrm{mm}.\;$ The maximum and minimum lengths of the air spring were calculated based on Equations (4) and (5), respectively. $L_{A1D1}\;$= 191.77 mm and $L_{A2D2}$ = 108.29 mm. Therefore, the JBF-165-200-2 bladder air spring was selected according to the bearing capacity and stroke. Its performance parameters are presented in Table 2.

The required damping coefficient of the damper was obtained according to the limit load [30]:(6)$$c=2\xi \sqrt{\frac{F_{Tmax}}{g}K}$$
where c denotes the equivalence damping coefficient (N·s/m) and K represents the stiffness of air spring (N/m). The air suspension system exhibits weak damping, and its relative damping ratio ξ is taken as 0.2−0.35. ξ was set as 0.3 and c  was suitable in the range of 725.13–1450.26 N·s/m. The customized 5-gear adjustable hydraulic damper produced by ARM company was selected, and the damping coefficients of each gear are presented in Table 3.

### 2.3. Comparative Analysis and Test on the Influence of Elastic Elements on the Dynamic Characteristics of Spray Boom Suspension

The transient and sinusoidal excitations were applied to the spray boom-air suspension system, and their dynamic response characteristics were analyzed. In addition, the spray boom-coil spring suspension and spray boom-rigid support systems were divided into two control groups. The spring and support were installed in the same position to keep the spray boom in a horizontal state when stationary. The specific procedure for applying transient excitation is presented as follows: (1) use the lifting method [21]; (2) apply a pulling force at the centroid of the spring boom; (3) adjust the spray boom to rotate forward by 8°; (4) remove the pulling force after the spray boom becomes stable. After that, an acceleration sensor recorded the acceleration change at the end of the spray boom with time.

The procedure for the applying vertical sinusoidal excitation is as follows: (1) apply vertical sinusoidal excitation using a mechanical vibration table; (2) use an acceleration sensor to measure and calculate the root mean square of the boom end acceleration (RMS1) and vibration table acceleration (RMS2), as well as the ratio of RMS1 and RMS2 when the spray boom is stable. Because the spray boom-air suspension belongs to passive suspension and could function as a low-pass filter, this paper mainly explored the dynamic characteristics of suspension in a low-frequency region. Meanwhile, to assess the influence of the resonance effect, the frequency area should include the natural frequency of the suspension and spray boom. Therefore, the selected excitation frequency was 0.1–3.5 Hz, and one group of tests was carried out every 0.2 Hz. Figure 4 presents the layout of test instruments. The rigid support primarily functions as support and connection; hence, it has a negligible damping effect. This test was only conducted for the spray boom-air and spray boom-coil spring suspensions to guarantee the test safety and avoid strong resonance.

### 2.4. Comparative Analysis and Test on the Influence of Different Soil Conditions

To explore the vibration suppression effect of spray boom-air suspension parameters under different soil conditions, in June 2020, field experiments in water and dry fields were conducted in the experimental field of the Daqiao community, Yongning Town, Nanjing City, China, as illustrated in Figure 5. Water and dry soils were selected for the field experiments. The soil moisture content was measured via the oven drying method [32,33], took the average of three measurements as a result. The sensor of the soil firmness measuring instrument (TJSD-750 type, Top Instrument, Zhejiang, China, the measuring depth: 0~375 mm, the measuring accuracy: ±1%, the cone opening angle: 30°) was inserted into the soil to measure the soil firmness and took the average of three measurements as a result. The specific values are presented in Table 4. The moisture content and firmness were measured three times, and the average value was taken.

With the volume of the additional air chamber increasing, the decreasing trend of the suspension dynamic stiffness slows down until the dynamic stiffness is stable [23]. Based on that, several small additional air chambers were selected, with volumes of 0.6 L, 1.2 L, 1.8 L, 2.4 L and 3.6 L. The damper adjusted the damping coefficient for the 5-gear transformation. A higher gear level indicated a smaller internal throttling area, a greater damping force generated under the same excitation, and a greater damping coefficient. When the sprayer was working, the half tank water was added to the medicine box to simulate the operation of the pesticide application, and the speed of the sprayer was controlled at 1 m/s. The acceleration data of the end of the spray boom sensor were collected during the smooth operation of the sprayer.

## 3. Results and Analysis

### 3.1. Comparison of Vibration Suppression Effects of Elastic Elements on the Spray Boom under the Transient Excitation

With the lifting method, the transient excitation was applied to the spray boom to obtain the time-domain response curve of the acceleration at the end of the spray boom, as illustrated in Figure 6. The rigidity of the spray boom suspension significantly influenced the acceleration of the spray boom end. In addition, the elastic element type exerted a significant effect on the vibration attenuation speed of spray boom; however, it had a negligible impact on the acceleration of the spray boom end. Compared with the coil spring, the air spring accelerated the stable speed of the spray boom by 1.1 times. The nonlinear stiffness of the air spring could better adapt to the spray boom vibration under the transient excitation, which improved the energy conversion efficiency of the system.

According to Table 5, compared with the spray boom-rigid support system, the spray boom-coil spring and spray boom-air suspension systems can reduce the peak vibration acceleration of the spray boom by more than 92%. In addition, these two systems can decrease the root mean square of the acceleration by over 81% and accelerate the vibration attenuation speed of the system by 2.5–5.3 times. Both the vibration time and frequency domains became smaller. Coil and air springs can reduce the system’s natural frequency of each order to keep the resonance of the spray boom in the high-frequency domain lower than 8 Hz. However, the risk of the spray boom resonance increased at approximately 1 Hz.

### 3.2. Comparison of Vibration Suppression Effects of Elastic Elements on the Spray Boom under the Sinusoidal Excitation

When sinusoidal excitation was applied to the spray boom, the frequency-domain response curve of RMS1 and RMS2 under different elastic elements are illustrated in Figure 7. Figure 8 presents the ratio of RMS1 and RMS2 under different elastic elements. When the sinusoidal excitation frequency was in the range of 0.3–3.5 Hz, the air spring kept the ratio of RMS1 and RMS2 less than 1, thus indicating an optimal vibration isolation effect. When the excitation frequency was 0.1 Hz, the air and coil springs amplified the vibration. The underlying cause was the slow spring deformation and negligible dynamic stiffness, which was not conducive to damping. Hence, the vibration suppression effect was not completely played. When the spray boom was equipped with the coil spring, the vibration became stronger at the sinusoidal excitation frequency of 1.1 Hz, and RMS1 was 2.8 times that of RMS2. This occurred because the sinusoidal excitation frequency was close to the system’s natural frequency. At this point, resonance was observed in the spray boom-coil spring system. Owing to the installation of an air spring, RMS1 did not exhibit a sudden increase trend, the main reason is that the air spring had ideal nonlinear characteristics, and its elastic characteristic is gentle and little change, thereby indicating that the resonance was weakened by the air spring under the sinusoidal excitation frequency.

### 3.3. Influence of Spray Boom-Air Suspension System on Damping Performance under Different Soil Conditions

According to Figure 9, during the operation in the water field, under five additional air chambers with volumes of 0.6, 1. 2, 1.8, 2.4 and 3.6 L, the damping coefficient of the damper had basically the same effect on the spray boom vibration, such that the vibration intensity of the spray boom decreased first, and then increased with the increase in the damping coefficient. When the damping coefficient was 1792 N·s/m (gear 4), the volume of each additional air chamber exhibited the best damping effect. With the increase in the damping coefficient and under the same excitation, the damping produced by the damper increased. Furthermore, the stiffness of the spray boom-air suspension system increased, and the spray boom vibration decreased. However, when the damping coefficient exceeded 1792 N·s/m, the stiffness of the suspension system was too large, which made the damping effect of the air spring unable to be fully prominent. In addition, the vibration of the machine body was prone to be transmitted to the spray boom.

Under the same damping coefficient, an additional air chamber with a smaller volume would generate a more evident vibration suppression of the suspension. The underlying reason is that the water field adopted had a low soil firmness, and the machine body had a small excitation amplitude and frequency [19]. The air suspension generated damping via the exchange of gas between the air spring and additional air chamber. The smaller the volume of the additional air chamber, the worse the smoothness of air flow movement, which is conducive to increasing the dynamic stiffness of the air spring and generating large damping to attenuate the vibration.

Hence, when operating in the low firm soil environment, the spray boom air suspension should match the additional air chamber with a small volume, and the damping coefficient of the damper should be approximately 1792 N·s/m.

The vibration of the spray boom during the operation in the dry field is illustrated in Figure 10. The effect of the damper damping coefficient on the spray boom vibration was the same as that in the water field operation. When other conditions remained unchanged, the increase in the damping coefficient triggered the increase in the suspension system stiffness and damping. The excessive damping of the damper influenced the vibration reduction in the air spring. Hence, under the volume of each additional air chamber, the vibration intensity of the spray boom initially decreased, and then increased with the increasing damping coefficient. However, the optimal damping coefficient in terms of the dry field was 1316 N·s/m, less than that under the water field condition. This is owing to the faster and larger frequency and amplitude of excitation in the dry field system. A smaller damping coefficient can produce sufficient damping to suppress vibration.

On the dry field, when the damping coefficient of the damper was the same, the vibration suppression of the suspension became more significant with the increase in the volume of the additional air chamber. The excitation with the large amplitude and fast frequency was applied to the spray boom, and the damper generated large damping. The air suspension needed a large additional air chamber to improve the smoothness of the air flow exchange between the spring airbag and additional air chamber. Accordingly, the vibration energy could be absorbed in time. A small additional air chamber with a large system stiffness was not conducive to vibration reduction. Therefore, when working in a solid soil environment, the spray boom air suspension should match the additional air chamber with a large volume [19], and the damping coefficient of the damper should be approximately 1316 N·s/m.

This study also investigated the specific relationship between the air spring and damper parameters of the suspension damping system. According to the experiment parameters, the fitting formula reflecting the relationship among the additional air chamber volume (*V*), damper damping coefficient (*c*), and acceleration root mean square (RMS) was obtained using MATLAB processing.

Water field conditions:(7)$$RMS=\frac{16.75+30.47V^{2}-8.25V^{3}-0.004c+0.00000625c^{2}}{1-0.83V+3.36V^{2}-0.85V^{3}+0.001c}$$

The square of the correlation coefficient R^2^ is 0.916061.

Dry field conditions:(8)$$RMS=\frac{4.36-2.30V+0.43V^{2}-0.0007c+0.0000001525c^{2}}{1-0.729V+0.192V^{2}-0.0138V^{3}-0.0000282c}$$

The square of the correlation coefficient R^2^ is 0.958849.

## 4. Discussion

The necessity of pesticide application and its effect improvement has been addressed for decades. As the most important crop in Asia, rice is a crop that requires a high frequency and considerable amount of pesticide. Thus, the study of pesticide application in rice is essential. As a fundamental way, the small-scale high-clearance sprayers are widely used in South China and South Asia, where the land is small.

Although the medium and small-scale high-clearance sprayers have the characteristics of mobility and flexibility, it is not easy to install complex spray boom damping systems due to the limit of load and power [27]. It has always been a question to attenuate the spray boom vibration of the medium and small-scale high-clearance sprayers in the absence of complex spray boom damping systems that cannot be installed [34,35].

In this study, a spray boom-air suspension system suitable for medium and small-scale high-clearance sprayers was designed, and the dynamic characteristics of the spray boom-air suspension system were analyzed. Moreover, the field experiments were carried out to study the influence of controllable parameters on the inhibition effect under different soil conditions. This study provides a new idea for vibration suppression of medium and small-scale high-clearance sprayers and improving droplet deposition distribution uniformity. Exploring the spray boom- air suspension system allows the farmers to maximally enhance the uniformity of droplet distribution and reduce the pesticides loss. The spray boom-air spring system can be a starting point to provide such a universal tool for small-scale high-clearance sprayer users. The limitations of the spray boom-air spring system were also identified for future developments.

Firstly, the adaptability range of this spray boom-air suspension system and the performance of the air suspension system in some extreme conditions (drainage ditch under rice rotation condition or uneven road surface) are not clear. The failure conditions or the maximum functional interval need to be further defined. Meanwhile, it is necessary to study further the quantitative relationship between the parameters (length and fulcrum of spray boom) and the design parameters of the air suspension to provide a general theoretical basis for the design of air suspension system of different spray boom.

Secondly, the current spray boom- air suspension system is still passive, and the stiffness and damping characteristics cannot be adjusted dynamically according to the sprayer driving conditions (the motion state of the sprayer and the road condition, etc.). The spray boom-air suspension system cannot always be in the best vibration reduction state. Therefore, researchers can combine sensor and control technology to achieve active suspension adjustment.

Despite the limitations, the spray boom-air spring system can provide a convenient method for suppressing spray boom vibration related to pesticide applications. Extending the spray boom-air spring system to intelligent spray applications may be feasible to enhance its accessibility, which will help farmers in ensuring spray quality and safe applications.

## 5. Conclusions

(1)A spray boom-air suspension system was designed. This paper described the composition and layout of the system. After theoretical calculations, the air spring and damper were selected to reduce the vibration intensity of the boom. This paper serves as a foundation for improving the spray uniformity of high-clearance sprayers.(2)Via a bench test, the dynamic characteristics of the spray boom suspension were analyzed. The obtained results indicate that under transient excitation, the spray boom equipped with the air spring can trigger the fastest vibration attenuation, the smallest peak value, and root mean square of the vibration acceleration. Under sinusoidal excitation, the air spring exhibited a better vibration suppression effect than the coil spring. The air spring expanded the vibration isolation frequency domain of the spray boom air suspension and weakened the resonance effect.(3)Field experiments were carried out to analyze the effects of soil conditions, additional air chamber volume of air spring, and damping coefficient of the damper on the vibration suppression of the spray boom-air suspension system. The obtained findings indicate that the soil with a low moisture content and high firmness was unfavorable to the vibration of the spray boom. When the soil firmness was low, the spray boom exhibited a low vibration frequency and intensity. The suspension system should include an additional air chamber with a volume of approximately 0.6 L, and the damping coefficient of the damper should be approximately 1792 N·s/m. Accordingly, sufficient damping can be generated. An excessively big additional air chamber with an over-large system stiffness was not suitable for reducing vibration. In addition, the excessive damping coefficient of the damper can potentially lead to an incomplete action of the air spring. Regarding the solid soil, the spray boom exhibited high vibration frequency and high strength. The suspension system should match a large additional air chamber of approximately 3.6 L, and the damping coefficient should be set to approximately 1316 N·s/m to avoid excessive stiffness, which was unsuitable for vibration damping. If the volume of the additional air chamber was too small and the damping coefficient of the shock absorber was too low, the suspension would not adapt to the system vibration. Furthermore, the large damper gear also triggered an incomplete action of the air spring.

## Figures and Tables

**Figure 1 sensors-21-06753-f001:**
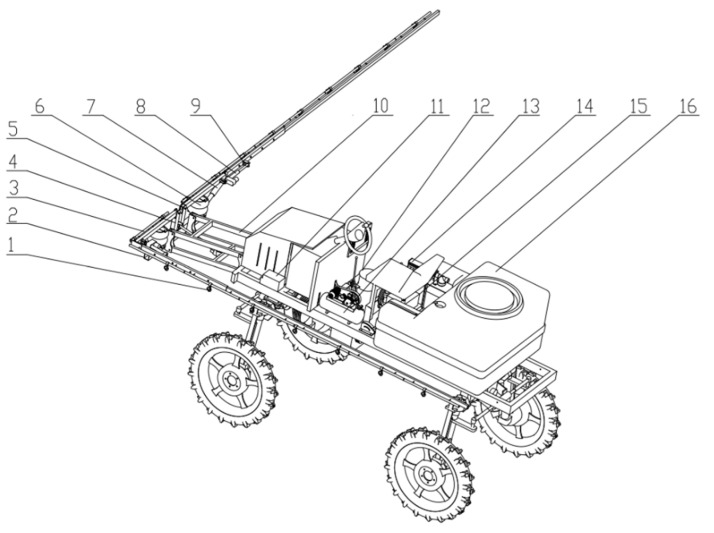
High-clearance sprayer structure 1. nozzle 2. suspension base 3. sprinkler 4. support bracket 5. pin shaft 6. air spring 7. damper 8. electric linear actuator 9. spray boom 10. parallelogram mechanism 11. accumulator 12. additional air chamber 13. air compressor 14. distributing valve 15. pesticide pump 16. pesticide box.

**Figure 2 sensors-21-06753-f002:**
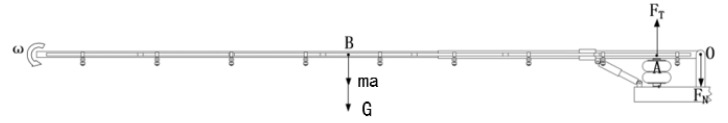
Free body diagram of spray boom.

**Figure 3 sensors-21-06753-f003:**
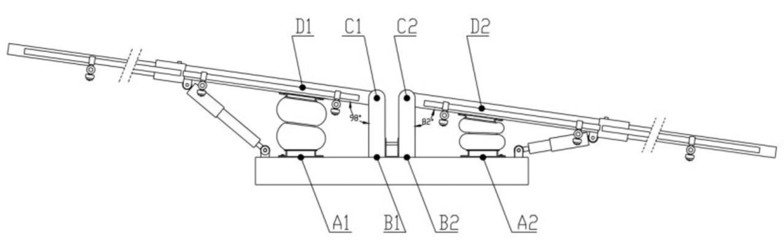
Operating diagram of air spring using the (A1, A2) the intersection point of air spring center line and base, (B1, B2) the intersection of fixed bracket vertical center line and base, (C1, C2) the center point of cross section of pin shaft, and (D1, D2) the intersection point of air spring center line and spray boom center line.

**Figure 4 sensors-21-06753-f004:**
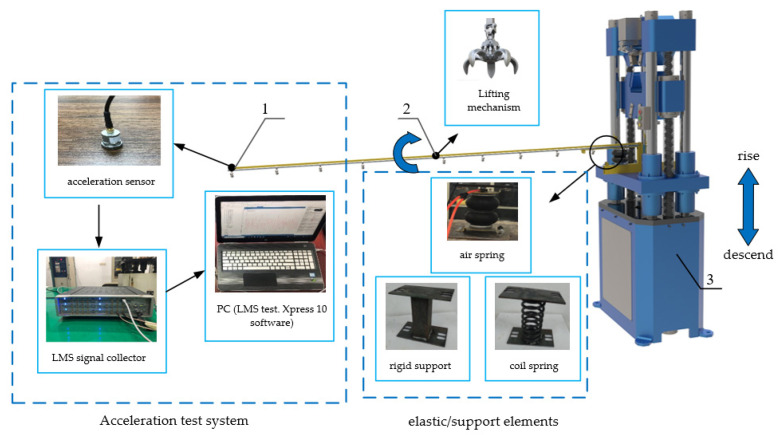
Layout of test instruments. 1. Measuring point of acceleration 2. Force application point of lifting mechanism 3. Vibration table.

**Figure 5 sensors-21-06753-f005:**
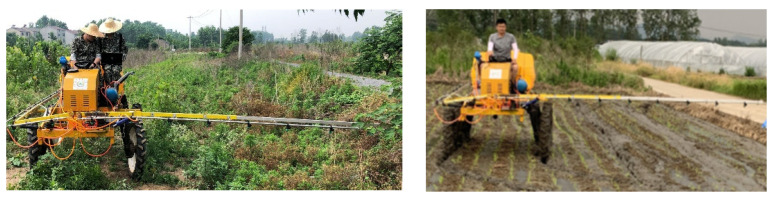
Field experiments.

**Figure 6 sensors-21-06753-f006:**
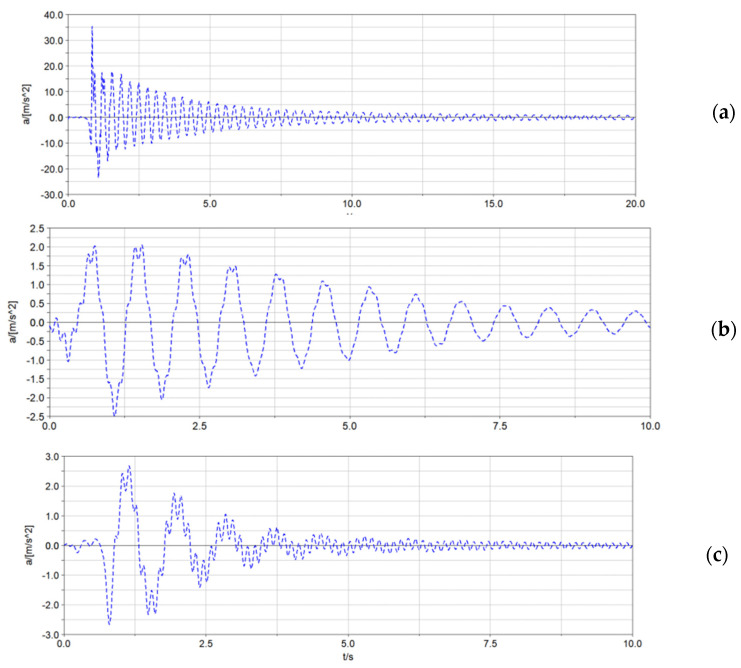
Time−domain response curve of the acceleration at the end of the spray boom. (**a**) spray boom-rigid support system; (**b**) spray boom-coil spring system; (**c**) spray boom-air spring system.

**Figure 7 sensors-21-06753-f007:**
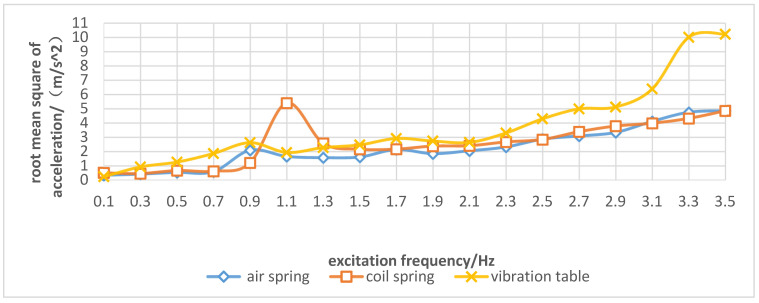
Frequency-domain response curve of root mean square of spray boom end acceleration (RMS1) and vibration table acceleration (RMS2) under different elastic elements.

**Figure 8 sensors-21-06753-f008:**
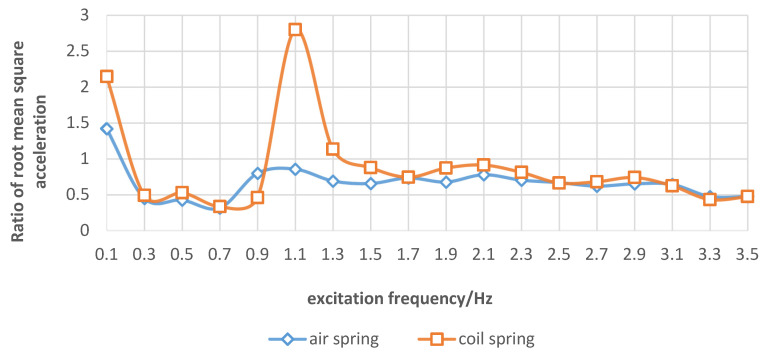
Ratio of root mean square of spray boom end acceleration and vibration table acceleration under different elastic elements.

**Figure 9 sensors-21-06753-f009:**
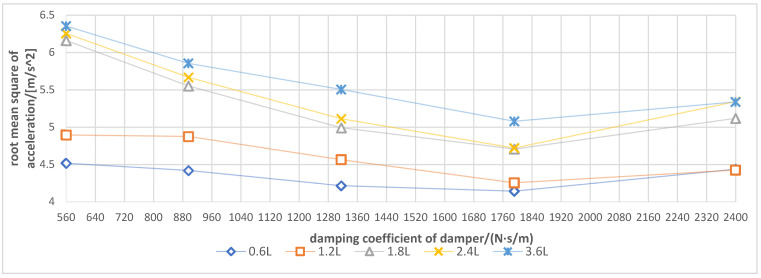
Response curve for root mean square of spray boom end acceleration under different damping coefficients and additional air chamber volume in the water field.

**Figure 10 sensors-21-06753-f010:**
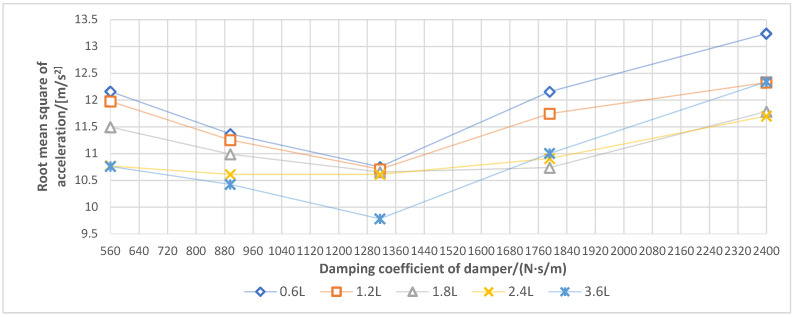
Response curve for root mean square of spray boom end acceleration under different damping coefficients and additional air chamber volume in the dry field.

**Table 1 sensors-21-06753-t001:** Principal parameters of medium and small-scale high-clearance sprayers and spray boom-suspension system.

Parameter	Value	Parameter	Value
Boundary dimension (length × width × height)/m × m × m	4.25 × 1.8 × 2.06	Maximum folding angle of spray boom/⁰	90°
Structural mass/kg	1150	Length of electric linear actuator/m	0.4
Power/kW	13.9	Length of suspension base/m	1
Operating speed/m·s^−1^	1–4	Length of single side spray boom/m	4.8
Wheel track (Between the left and right wheels)/m	1.5	Mass of single side spray boom and connector/kg	13
Spray height/m	0.48–1.48	Air compressor type	Positive displacement

**Table 2 sensors-21-06753-t002:** Performance parameters of air spring.

Parameters	Value
Maximum bearing capacity of spring (N)	7000
Natural diameter of spring (mm)	146
Natural height of spring (mm)	160
Maximum elongation of spring (mm)	200
Minimum compression of spring (mm)	75
Spring stroke (mm)	125
Maximum spring stiffness (N/mm)	176

**Table 3 sensors-21-06753-t003:** Damper parameters.

Gear	1	2	3	4	5
Damping coefficient/(N·s/m)	560	896	1316	1792	2400

**Table 4 sensors-21-06753-t004:** Soil firmness and moisture content.

Soil Condition	Soil Depth/cm	Moisture Content/%	Firmness/kPa
Dry field	5	2.47 ± 0.09	902 ± 5.00
	10	1.96 ± 0.08	1103 ± 6.56
	15	1.64 ± 0.04	1373 ± 6.08
Water field	5	22.07 ± 1.19	46 ± 3.61
	10	19.01 ± 0.47	158 ± 11.53
	15	14.87 ± 0.02	947 ± 5.57

**Table 5 sensors-21-06753-t005:** Boom transient response parameters.

Type	Peak Value/m/s^2^	Peak Time/s	Root Mean Square/m/s^2^	Stabilization Time/s	First-Order Natural Frequency/Hz	Second-Order Natural Frequency/Hz
Spray boom-rigid support system	35.47	0.85	4.29	16.42	3.13	11.72
Spray boom-coil spring system	2.49	1.10	0.83	7.24	1.37	7.62
Spray boom-air spring system	2.68	1.15	0.62	4.07	1.17	7.62

## Data Availability

Data sharing not applicable.

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
