# Peer review of "Analysis of Factors Influencing Vibration Suppression of Spray Boom-Air Suspension for Medium and Small-Scale High-Clearance Sprayers"

_sensors, 2021, doi:10.3390/s21206753_

Round 1

Reviewer 1 Report

As it appears from studying the article, it is quite interesting in general and addresses to the readership of the Journal. In particular the authors present the design of a spray boom-air suspension system. The paper describes the composition and layout of the system. Via a bench test, the dynamic characteristics of the spray boom suspension were analysed and some field experiments were carried out to analyze the effects of soil conditions, additional air chamber volume of air spring, and damping coefficient of the damper on the vibration suppression of the spray boom-air suspension system.

Some particular comments and suggestions for the authors so as to improve their manuscript are as follows:

  1. Although the references included in this work are appropriate, these are presented in a rather generic way in the introduction section. In order for the authors to substantiate the novelty of the proposed solution they are suggested to more thoroughly compare it with other similar solutions found in literature, even by adding more relevant references.
  2. The methodology followed in this research is described sufficiently enough and is supported by a thorough experimental analysis as well as by an efficient number of adequately illustrated figures, tables and charts, permitting other researchers to reproduce certain aspects of this research work.
  3. The results of the research are adequately presented and analysed while the conclusions drawn seem to be consistent with the evidence and arguments that were presented. However, a Discussion section is suggested to be added in order for the authors to discuss the results and how they can be interpreted in perspective of previous studies and of the working hypotheses. The findings and their implications should be discussed in the broadest context possible and limitations of the work highlighted. Future research directions may also be mentioned. This section may be combined with Results.
  4. Finally, the paper is in general well-structured and written in appropriate and understandable English language according to the standards of the Journal. However, some minor spellcheck might be needed.

Reviewer 2 Report

The paper compares spray boom suspension using various elastic elements: (a) rigid support system; (b) coil-spring system; and (c) air-spring system. Additionally, field experiments are used to evaluate the impact of soil conditions. The paper is interesting and well-written. I have only a few comments, as follows:

  • L162 Is there any reference for a 8 degree limit to the deflection angle?
  • L238 I think you must be referring to Table 4 here.
  • L245 Define the term "resonance risk". 
  • L284 why was resonance not observed in case of air-spring system?
  • L296-L317 This part could be placed under the Materials and methods section.
  • L302 Here, too, Table 4 is misleading.

Reviewer 3 Report

General thoughts

The article presented for review is correctly written and interesting in its subject matter. The article layout is correct, with the recommended corrections. I would also suggest supplementing the content of the article.

Detailed comments

Abstract - spelled correctly

The "Introduction" is relatively short, but it contains comprehensive information on the issue under consideration. The aim of the work I would suggest to formulate less descriptive, and more hypotheses to be proved or tasks to be achieved.

Chapter "Material and methods" written correctly. I would suggest clarifying the term "Wheel track" in Table 1 because it is not clear to me what the authors mean by this term. In Figure 2, the force "m‧a" is missing, also in the description the designation "m‧a" should be added. This chapter lacked information about the number of repetitions of measurements during the performance of individual tests.

Results and analysis chapter written correctly. I would propose a fragment concerning the methodological description of the field tests performed (lines 296 to 303) and move to the "Materials and methods" chapter. This fragment also lacks information on how many moisture content measurements were made, as well as how many measurements were made of the penetration resistance measured by the TJSD -750 device. It is worth adding what dimensions the penetrating cone had, especially when it comes to the cone opening angle. It would also be good to specify the standard deviation of the values ​​of moisture and soil penetration resistance (Table 5)

The chapter Conclusion does not raise any objections. The question that arises is whether the authors propose variable volumetric dimensions of the additional pressure accumulator and the possibility of changing the characteristics of the Damper in the actual technical solution of the system

Kind regards
